# scafSLICR: A MATLAB-based slicing algorithm to enable 3D-printing of tissue engineering scaffolds with heterogeneous porous microarchitecture

Ethan Nyberg[1,2☉], Aine O'Sullivan[1,2☉], Warren Grayson[1,2,3,4] *

**1** Translational Tissue Engineering Center, Johns Hopkins University School of Medicine, Baltimore, Maryland, **2** Department of Biomedical Engineering, Johns Hopkins University School of Medicine, Baltimore, Maryland, **3** Department of Materials Science and Engineering, Johns Hopkins University, Baltimore, Maryland, **4** Institute for NanoBioTechnology, Johns Hopkins University, Baltimore, Maryland

☉ These authors contributed equally to this work.
* wgrayson@jhmi.edu

**Data Availability Statement:** All relevant data are within the manuscript and its Supporting Information files.

## Abstract

3D-printing is a powerful manufacturing tool that can create precise microscale architectures across macroscale geometries. Within biomedical research, 3D-printing of various materials has been used to fabricate rigid scaffolds for cell and tissue engineering constructs with precise microarchitecture to direct cell behavior and macroscale geometry provides patient specificity. While 3D-printing hardware has become low-cost due to modeling and rapid prototyping applications, there is no common paradigm or platform for the controlled design and manufacture of 3D-printed constructs for tissue engineering. Specifically, controlling the tissue engineering features of pore size, porosity, and pore arrangement is difficult using currently available software. We have developed a MATLAB approach termed **scafSLICR** to design and manufacture tissue-engineered scaffolds with precise microarchitecture and with simple options to enable spatially patterned pore properties. Using scafSLICR, we designed, manufactured, and characterized porous scaffolds in acrylonitrile butadiene styrene with a variety of pore sizes, porosities, and gradients. We found that transitions between different porous regions maintained an open, connected porous network without compromising mechanical integrity. Further, we demonstrated the usefulness of scafSLICR in patterning different porous designs throughout large anatomic shapes and in preparing craniofacial tissue engineering bone scaffolds. Finally, scafSLICR is distributed as open-source MATLAB scripts and as a stand-alone graphical interface.

## Introduction

3D-printing technologies have become widely available with a large number of commercially available low-cost hardware systems and printable materials used to fabricate scaffolds for tissue regeneration [1,2,11–13,3–10]. Thus far, much of the work using fused deposition

**Funding:** This work was supported by the Office of the Assistant Secretary of Defense for Health Affairs, through the Vision Research Program Technology/Therapeutic Development Award under Award No. W81XWH-15-VRP-TTDA (W.G). Opinions, interpretations, conclusions and recommendations are those of the author and are not necessarily endorsed by the Department of Defense (W81XWH-16-1-0758). The funders had no role in study design, data collection and analysis, decision to publish, or preparation of the manuscript.

**Competing interests:** The authors have declared that no competing interests exist.

modeling (FDM) to produce constructs for tissue engineering has focused on developing suitable biomaterials with a variety of mechanical and biological properties [14]. Scaffolds for tissue engineering necessarily contain highly-defined porous networks. The interplay between scaffold mechanics and porous volumes for uniform cell seeding, promoting *de novo* tissue growth, and the diffusion of nutrients throughout presents a challenge in determining the optimal microarchitectural design for a scaffold. While there have been design approaches for selective laser sintering 3D-printing using unit cell libraries [15], topology optimization [16,17], and mathematical design [18], the structures are not reasonably transformed to the fiber deposition paradigm of FDM [19].

An ongoing tissue engineering design goal is to create spatially controlled, heterogeneous patterns of pores throughout anatomic shapes to mimic differences in mechanical requirements throughout the tissue. In prior studies, this has most commonly been achieved by modifying the fiber-fiber spacing between *z*-levels, which results in different pore sizes at different heights in the print. For example, Sobral *et al.* designed a gradient of pore sizes by systemically increasing or decreasing fiber-fiber spacing in each print layer [20]. Additionally, Woodfield *et al.* 3D-printed cartilage constructs with a gradient of pore sizes in the *z*-direction [21]. These fiber-spacing approaches are constant across an xy-plane and limit designs to changes in spacing in the *z*-direction. While such an approach is applicable in small-scale prints, it does not easily transfer to human-scale complex anatomic shapes. Di Luca *et al.* demonstrated fiber-fiber spacing across the *xy*-plane, resulting in a step gradient across the plane [22–24]. Therefore, the fiber-fiber spacing could be controlled in *xy* and *z* directions simultaneously, enabling designs with different pore sizes across 3D-space. Thus far, this approach has only been shown on a small scale, cuboid scaffold with a three-pattern linear gradient. Implementation of such gradients in the *xy* plane across a variety of large, more complex geometries remains an unmet challenge for bone tissue engineering.

Additionally, specific control over pore architecture is desirable. Fiber height is often mismatched from desired pore sizes, and cross-hatching fiber patterns on alternating print layers result in pore diameters that are determined by the versatile fiber-fiber spacing in the *z*-direction but limited by fiber height in the *xy*-direction. Varying the fiber height can change the height of the pores in the *xy*-plane, but this approach must be implemented across the entire print plane which prevents in-plane patterning. Further, fiber height is limited by the range of nozzle hardware. However, repeating the same print pattern without changing the fiber location could result in stacked, taller fibers and therefore pores, as shown by Moroni *et. al.* [25] and Xu *et. al.* [26]. Struts designed from congruent, stacked fibers could be used to make pore microarchitectures isotropic and vary pore size in all three spatial dimensions.

The process of transforming a 3D-design into the *xy*-layers of hot element paths is termed *slicing*. The slicing process determines the fiber laydown pattern, and the resulting geometric and mechanical properties of the print [27]. Traditional slicing software systems create a solid wall or shell around the exterior surface of the shape with a single, infilling, truss pattern applied to the interior bulk of the shape. These resulting designs are not useful for tissue engineering constructs as they do not contain interconnected pore networks. Research groups have been limited by the set of functionalities in broadly-used software (such as Slic3r [28] or Cura [29]) or in the proprietary software delivered with the bioprinter–which has restricted the availability of useful tissue engineering designs.

To overcome the limitations of the available slicing software, several groups have prepared custom porous designs through 'brute force' design: i.e. they manually design each pore and strut in CAD programs and then pass the CAD file to a traditional slicing program, which best approximates fiber placement across the design [5,30]. This design process is labor and computationally intensive and disconnects the design process from the design space of the 3D-

printer, which can cause infidelities in the final product. Alternatively, custom slicing software can enable the creation of gradients or custom porous structures. For example, Kang *et*. *al*. developed an integrated tissue-organ printer and custom software integrated into the system to design and manufacture of constructs. However, their published source code is unique to their hardware and does not appear to allow for the design of gradients [9]. Trachtenberg *et al*. developed a Python and Pronterface system to generate GCODE that can vary the fiber-fiber spacing on different print levels on a custom-built 3D-printer [31]. However, these programs are specialized to each design and manufacturing system and are not easily replicable or adaptable.

Therefore, currently available slicing programs do not allow precise control over porous patterns and microarchitectural features needed for tissue engineering scaffolds. Here, we present an approach to designing 3D-printed scaffolds with patterns of porous and mechanical properties. The goals of this approach are to (i) develop a software which can design and implement patterns of pore properties throughout 3D space which contain isotropic, fully connected pores relevant to tissue engineering, (ii) validate the printability and mechanical integrity of such designs, and (iii) provide this software as a tool that researchers can use when 3D-printing tissue engineering scaffolds. Additionally, we demonstrate that the resulting approach allows the independent patterning of pore size and porosity across a variety of anatomic shapes relevant to craniofacial bone regeneration.

## Materials & methods

### 3D-printing on Lulzbot

The methods in this paper were developed for use on a Lulzbot Taz 5 3D-Printer (Aleph Objects, Loveland, CO), which is representative of the many low-cost desktop 3D-printers that are broadly in use. The printer uses gears to drive a solid polymer filament through a melt chamber and narrow extruder nozzle. The nozzle is moved in the *x* and *y* directions as it deposits material in a single *z* level before proceeding in a layer-by-layer fashion until the build is complete. The cooling and solidification rate of the extruded polymer is critical for determining print quality, and it is controlled by adjusting air fans and the heat of the print surface. This paper uses the following terms to describe the 3D-printed part:

- **Fiber**–the structure of extruded material from the extruder head along a toolpath on a single print layer. Fibers are assumed to be rectangular with the width of the extruder nozzle and the height of the print layer.

- **Strut**–the solid material resulting from a set of adjacent fibers, often composed of multiple fibers in width and height.

- **Pore**–the channel-like void spacing between struts, in horizontal and vertical directions. Pores have square projections with equal width and height when viewed from the top or side of the scaffold.

Most importantly, the Lulzbot uses the Marlin operating system to process the standard RepRap flavor of GCODE instructions to control the robotic behavior of the system. The machine responds to commands to deposit a fiber of the material (*extruder diameter*) at a given temperature (*extruder temperature*), at a given rate (*extrusion rate*), and move in x-y space (*tool paths*, *extruder movement speed*). Additionally, the print surface can be heated to prevent warping (*bed temperature*) and the print can be cooled by turning the fan on at various print heights (*fan speed*, *fan start height*).

Scaffolds in this paper were printed using acrylonitrile butadiene styrene (ABS) plastic filament (IC3D, Columbus, OH) with the following printer settings. A 0.5mm diameter extruder was used with the extruder temperature set to 240˚C and the bed temperature set to 110˚C. The layer height was set to 0.2mm and fans were turned on to 50% speed after the first layer was deposited. The feed rate was set to 1200 mm/min and a 1% over-extrusion factor was applied throughout the entire scaffold. scafSLICR could work with a range of materials in an FDM printer, optimized with printing parameters.

## Scaffold design with MATLAB Script

The code was written to enable the feature-driven design of tissue engineering scaffolds. The key features of porosity and pore size are used to create a 3D-template, which is then applied to the desired areas of the scaffold shape (Fig 1). Pores are designed as isotropic square pores. Porosity is tuned by both the pore width and the strut width (design porosity, Eq 1). The strut width can be increased by placing multiple fibers directly adjacent to each other (fiber-fiber spacing = 0 mm) and pore width is controlled by the strut-strut distance. By repeating the same strut pattern on consecutive layers, the strut height can be increased to equal the pore width and result in square pores. The porosity of the repeating unit is calculated in Eq 1: a

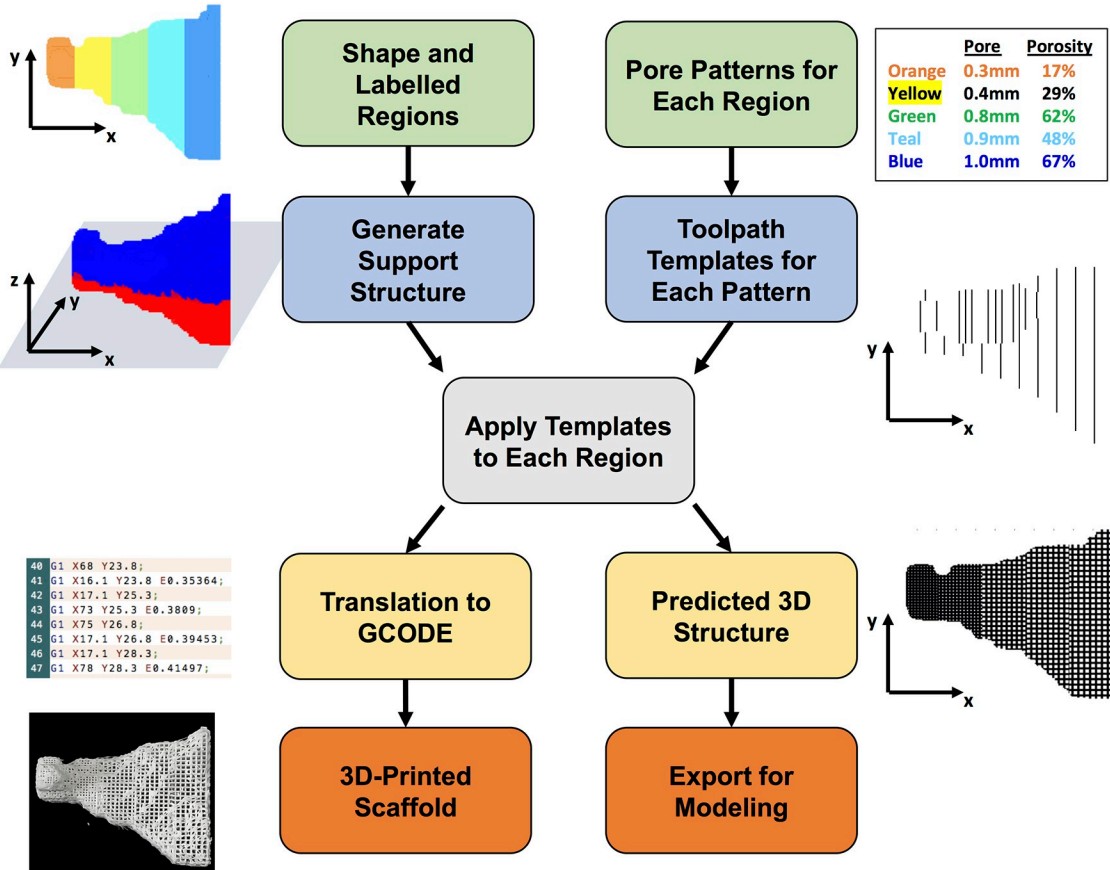

**Fig 1. Overview of scafSLICR approach.** User inputs a labeled 3D shaped and the pore properties for each label (green boxes). The program then generates a support structure between the shape and the print bed (blue/red shape) and tool path templates for each pore pattern (blue boxes). The slicing process convolves these tool path templates with each x-y level of the shape according to the label (gray box). The result of this convolution is then translated into a set of GCODE instructions or into a predicted porous model of the shape (yellow boxes). These outputs can be manufactured on a 3D-printer or used for *in silico* modeling (orange boxes).

repeating unit has the width of a strut + pore, the length of a strut + pore, and the height of pore, while the void volume is given by pore width*pore width*pore height.

$$porosity = \frac{(pore\ width)^\wedge 2}{(pore\ width + strut\ width)^\wedge 2} \tag{1}$$

The inputs to the code are 3D-shape, pore sizes, and porosities. The possible combinations of pore sizes and associated porosities are dependent on the thickness of the struts and are available as design options. The shape can be easily adapted from CT scans, STL files, or other 3D-data. The pore size and porosity can be inferred from the initial biologic data (e.g. CT density), mathematically defined in a variety of gradients, or any desired pattern that can be applied to a 3D-matrix.

The scafSLICR function generates templates of toolpaths in both *x* and *y* directions based on these parameters and convolves them with the shape matrix. Additionally, it generates support material between the input shape and the print bed surface. The function includes the options to improve print quality by pausing, back-tracking, or retracting material at the end of each fiber to prevent dragging strands across pore spaces.

The program outputs include common GCODE instructions that are conserved across many common FDM (tested on the RepRap Marlin system [32]) printers and 3D rendering of expected design (as STL and volumetric data). For ease-of-use, the function was incorporated into a graphical user interface [S1 Appendix]. It uses 3D-plotting [33] and STL import [34] scripts from the Mathworks repository.

scafSLICR is freely distributed on the Mathworks repository at www.mathworks.com/matlabcentral/fileexchange/72856-scafslicr as well as in supplements to this article.

## Scaffold manufacturing

Scaffolds were manufactured to assess the print quality of different porous patterns (homogenous scaffolds), the transition between different patterns (hybrid scaffolds), and gradients of patterns in three dimensions (gradient scaffolds). Homogenous and hybrid scaffolds were $20 \times 20 \times 10$ mm and gradient scaffolds were $30 \times 30 \times 30$ mm. The exact porous features of all scaffold groups are listed in Table 1. 3D-Design models were generated by scafSLICR by assuming fibers to be perfectly rectangular (nozzle width x print layer height). The design was assembled into a volumetric 3D-matrix which could be examined directly in MATLAB using matrix property analyses, MATLAB 3D-plotting functions, or exported as an STL to be viewed and analyzed in a range of software programs.

**Table 1. Pore features of homogenous, biphasic, and gradient scaffolds.**

| Homogenous | | Hybrid | | Gradient | |
|---|---|---|---|---|---|
| Pore Size (mm) | Porosity (%) | Pore Size (mm) | Porosity (%) | Pore Size (mm) | Average Porosity (%) |
| 0.2 | 28% | 0.2 → 0.5 | 28% → 25% | 0.2 | 28% |
| 0.5 | 25% | 0.5 → 0.8 | 25% → 28% | 0.35 | 26% |
| 0.8 | 28% | 0.2 → 0.8 | 28% → 28% | 0.5 | 50% |
| | 45% | 0.8 → 0.8 | 28% → 45% | 0.65 | 56% |
| | 62% | 0.8 → 0.8 | 45% → 62% | 0.8 | 62% |
| | | 0.8 → 0.8 | 28% → 62% | | |

## Print quality assessment

Design features were measured in manufactured scaffolds and evaluated for accuracy compared to the input values. scafSLICR was used to design and print scaffolds ($20 \times 20 \times 10$mm) with a variety of combinations of pore size and porosity, along with a solid ABS cube. The porosity of printed porous scaffolds was determined by mass measurements compared to solid prints of the same dimensions (Eq 2), and they were compared to the porosity of the computer design by computing the void fraction the 3D-MATLAB matrix.

$$Porosity_{mass} = 1 - \frac{mass_{porous}}{mass_{solid}} \tag{2}$$

Alternative Eq 2:

$$Porosity_{mass} = 1 - \frac{mass_{porous}/volume_{porous}}{mass_{solid}/volume_{solid}}$$

Scaffolds were imaged on a stereoscope (Zeiss Z8). Images were taken of top and side views at 2X magnification. Pore size and strut width were measured separately for top and side views. Pore size was analyzed using the DiameterJ plug-in for FIJI [35,36] by measuring the area of each pore of the binarized image. The size of the pore was then reported as the square root of the pore area. All of the pores were measured in each scaffold and each scaffold design was printed in triplicate. Strut widths were measured by hand in FIJI using the original stereoscope image. Between 59 and 69 struts were measured over three scaffolds per group. For pore size and strut width, the ratio of the measured value to the predicted value reported +/- standard deviation.

## Mechanical testing

Scaffolds were tested to assess the base mechanical properties of homogenous and hybrid scaffolds. Scaffolds measuring $20 \times 20 \times 10$ mm were loaded into an MTS Criterion Model 43 (Eden Prairie, MN) with a 5 kN load cell and subjected to unconfined uniaxial compression. The scaffolds were compressed perpendicular to the print axis at a rate of 1.27 mm/min. The compressive modulus was determined from the linear region of the stress-strain curve (n = 3).

## Analysis of porous boundaries

In order to assess the connectivity of pores between regions of different porous microarchitectures, the area of the interface was analyzed in the digital model in MATLAB. The interface was isolated digitally at a 200μm thickness, and the area of each connecting pore was measured. The porous area fraction of the boundary surface was found by summing the individual pore areas and dividing by the area of the interface boundary between regions.

## Anatomic shapes

Large portions of the craniofacial skeleton were selected to serve as anatomic test shapes. STLs or DICOMs of the shapes were exported from MIMICs (Materialise, Plymouth, MI) and imported into scafSCLICR. The shapes were divided into regions arranged linearly along the length of the shape (zygoma), or according to shape thickness (orbital bones), or according to depth (hemimandible). Different porous patterns appropriate for tissue engineering were selected from the design space and applied to the different regions of the anatomic shapes.

## Results

scafSLICR was used on a standard desktop computer to generate the designs in this study [S2 Appendix, Examples 1–9]. The largest shape (orbital bone) took six minutes to slice and generate the GCODE file, which is similar to the computing time when using Slic3r. Smaller volume shapes required proportionally less time to slice. 3D-printing time similarly scaled with volume, with large shapes requiring on the order of 10 hours and smaller shapes on the order of 20 min. Designed scaffolds were manufactured using the output GCODE without complications. Isotropic, regular, cubic pores were visible from top-down and side-on views of the scaffold (Fig 2A). The support material was automatically generated for anatomic shapes and removed from prints with minor artifacts.

### Available design space

Based on the diameter of the printer nozzle in use, the strut width can be modulated by depositing adjacent fibers (Fig 2A), thus a variety of strut widths may be achieved that are integer multiples of the printer nozzle diameter. This allows the decoupling of pore diameter and overall porosity. By modulating the strut width, a multitude of porosities may be achieved for a given pore diameter (and *vice versa*) as shown in Fig 2B. Pore diameters ranging from 0.2 mm to 1.0 mm were successfully printed using the 0.5 mm nozzle on the Lulzbot Taz5 printer. Within this range of pore diameter, many different porosities may be achieved by varying the strut width. For example, for a pore diameter of 500 μm, eight different porosities may be achieved between 11% and 50% by increasing the strut width from 0.5 mm to 4 mm.

The maximum porosity is determined by the pore size and printer nozzle diameter. The maximum porosity for pore diameters ranging from 0.2 mm to 1.0 mm is summarized in Table 2. For a 0.5 mm nozzle diameter, porosities may be achieved between 29% and 67%, and the porosity may be further increased to 74% by using a printer nozzle with a diameter of 0.35 mm.

Similarly, a specific porosity may be achieved using multiple different pore diameters. A porosity of 28.57% can be achieved at 0.2, 0.4, 0.6, 0.8, and 1.0mm pores by modulating the strut width between 0.5mm and 2.5mm (Fig 2B).

### Print quality/validation of predicted designs

This study evaluated five combinations of pore size and porosity that were considered to be highly relevant for bone tissue engineering applications (summarized in Table 2). These designs were printed and used to validate that the predicted designs from scafSLICR could be successfully manufactured with a high degree of fidelity. Printed scaffolds are shown with their respective design previews in Fig 3A. Manufactured scaffolds matched predicted designs to a high degree in both the top and side views. There was slight over-deposition of material, with pore sizes consistently below the predicted value irrespective of the actual pore diameter (Fig 3B and 3C). Pore diameter ranged from 76% to 93% of the expected value while the strut width varied from 3.5% under deposition to 13% over deposition.

The measured gravimetric porosity (Fig 3D) is strongly correlated to the specified porosity of input design. The deviation of measured porosity from input porosity is due to the dimensions of the printed scaffold not being exact multiples of the characteristic distances of the individual microarchitectures (pore width in $z$, pore width + strut width in $x$ and $y$).

Homogenous scaffolds were compressed to find the effective compressive modulus (Fig 3E). Primarily, the effective compressive modulus decreased with increased porosity. Increasing the porosity from a solid cube to 28% porosity with 200 μm pores resulted in a 44% decrease in compressive modulus. Further increasing the porosity to 62% with 800 μm

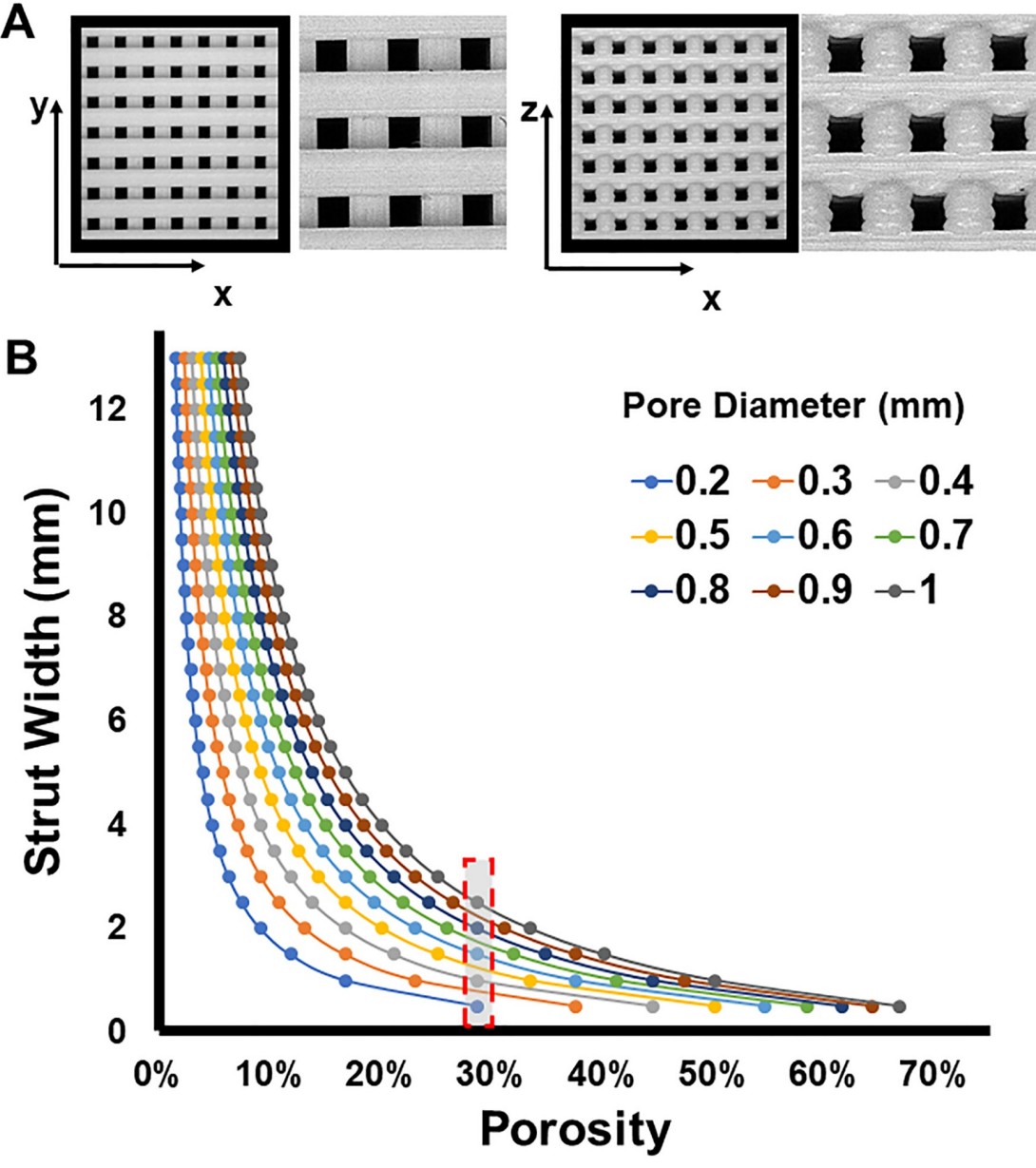

**Fig 2. Available design space.** (**A**) Stereoscope pictures (1X, 5X) of scaffolds produced with scafSLICR demonstrating isometric pores. (**B**) Relationship of Strut Width and Porosity: Modulating the width of struts can produce a range of discrete porosities that are manufacturable at a given pore diameter for 0.5 mm nozzle.

pores resulted in an 84% decrease in compressive modulus relative to the solid cube. Despite the clear inverse relationship between porosity and compressive modulus, the three designs with near 25% porosity had different compressive moduli, demonstrating that the mechanics also vary with the specific microarchitecture (pore size and strut size). Increasing pore size also resulted in decreased modulus with scaffolds containing 200, 500, and 800 μm pores with near 28% porosity having a compressive modulus of 503, 486, 328 MPa, respectively.

**Table 2. Maximum porosity for a range of pore diameters.**

| Pore Size | Upper Porosity Limit (0.5mm nozzle) | Upper Porosity Limit (0.35mm nozzle) |
|---|---|---|
| 0.2mm | 29% | 36% |
| 0.3mm | 38% | 46% |
| 0.4mm | 44% | 53% |
| 0.5mm | 50% | 59% |
| 0.6mm | 55% | 63% |
| 0.7mm | 58% | 67% |
| 0.8mm | 62% | 70% |
| 0.9mm | 64% | 72% |
| 1.0mm | 67% | 74% |

## Hybrid scaffolds

The porous interconnectivity between different microarchitectures in hybrid scaffolds (Fig 4A) was analyzed in predicted designs (Fig 4C). Because the nature of the interface depends on the position, extent, and curvature of the interface surface, pattern-to-pattern interconnectivity could not be assessed experimentally and was instead measured using *in silico* designs of the presented examples. All interface designs included connected pores. A portion of the connected pores was often reduced in individual area, but together represent a large area fraction of connected porous space (10–30% of boundary area) per interface. Hybrid scaffolds were also tested for mechanics in compression normal to the plane of transition between microarchitectures (Fig 4B and 4D). The modulus of the hybrid scaffold was compared to the modulus of the more porous (softer) design and less porous (stiffer) design. In all cases, the hybrid scaffolds had moduli between those of the two constitutive homogenous designs. This result indicates that the transition between microarchitectures did not weaken the mechanics of the scaffold.

## 3D printed scaffolds with heterogeneous porous patterns

Gradient patterns of different porous microarchitectures were applied to cubes (Fig 5). First, we demonstrate the ability of scafSLICR to prepare gradients. It readily applied gradients in the print (*z*) direction (Fig 5A) or across the print layer (*xy* plane) (Fig 5B). Further, a 3D gradient was applied which graded the porous microarchitectures from the exterior to the interior of the cube (Fig 5C). The cubes were larger than homogenous or hybrid scaffolds in order to accommodate the characteristic sizes (twice the sum of the pore and strut width) of the five patterns. The designs were 3D-printed without complication.

Portions of the craniofacial skeleton were used to test shape complexity, pattern complexity, and scale. The zygomatic bone (Fig 6A) was printed with a linear gradient in the pore structures from left to right, arranged so the less porous design was at the narrow portion of the bone and the more porous was at the wider portion of the bone.

The hemi-mandible (Fig 6B) was graded into shells based on depth from the surface of the shape. A more porous pattern was applied to the outer shell, versus a more solid pattern along the inner core. This shell design could allow for cell ingrowth into the scaffold along the surface with some added stability from the inner core. There is satisfactory porous connection between the outer two shells, however, the inner core was nearly solid and did not have many pores for connectivity.

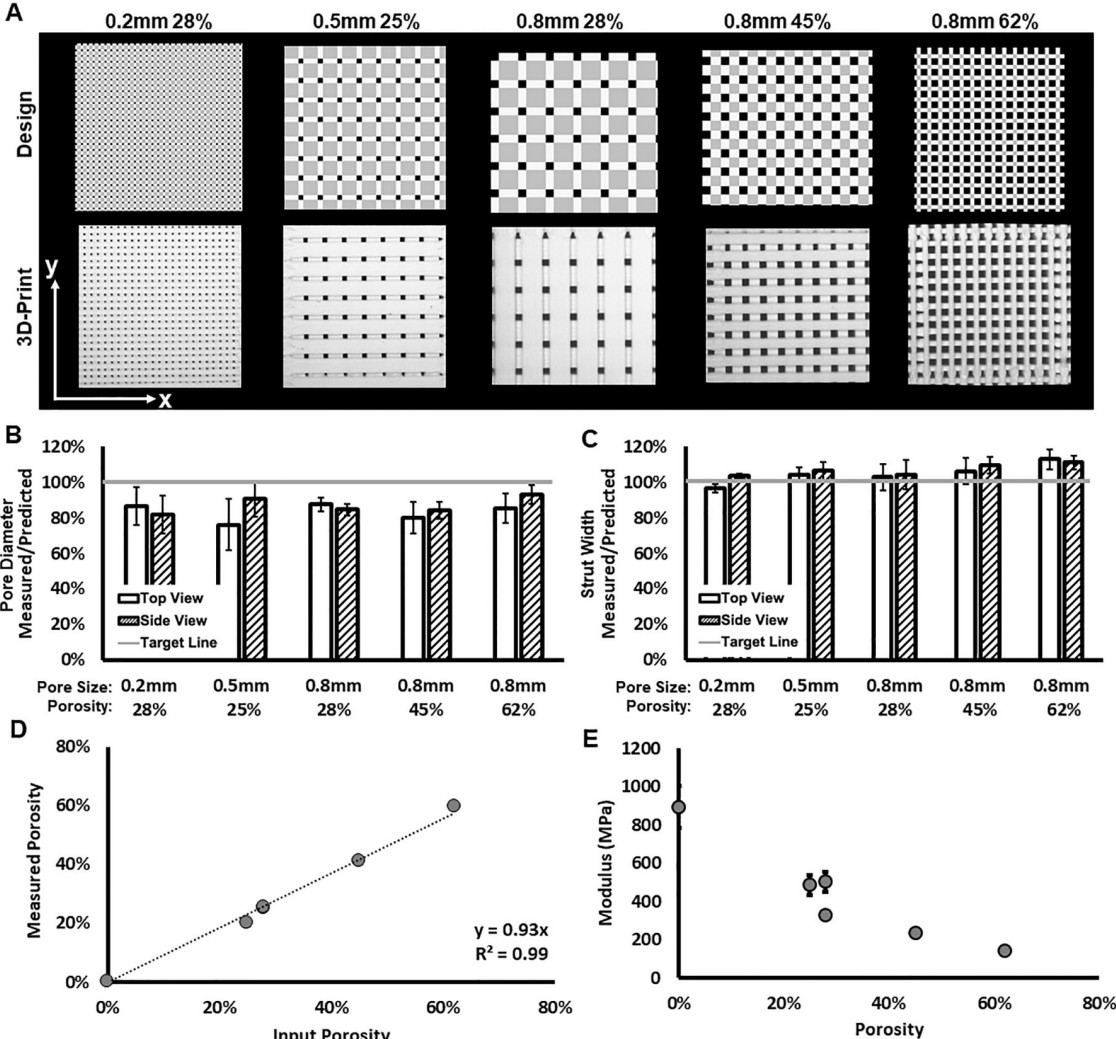

**Fig 3. 3D-printed scaffolds with uniform isotropic pores.** (**A**) Side-by-side comparison of scaffold previews (top row) and 3D-printed scaffolds (bottom row) for different patterns of pore size and porosity. (**B, C**) Assessments of print fidelity of pore diameter and strut width to design from top and side views. (**D**) Observed gravimetric porosity and expected design values. (**E**) Compressive modulus varies with porosity.

3D-printing thin structures is difficult, more so when the structure is manufactured with a porous pattern. The orbital bone shape (Fig 6C) has characteristically thin bones across the orbital floor. To print these faithfully, the shape was divided into regions based on the average thickness, which allowed the thin regions to be assigned a less porous, more stable pattern. Thicker, more stable regions were assigned more porous patterns. The arrangement of the patterns resulted in curved and interwoven boundaries throughout the shape. These boundaries maintained 10% and 20% area pore-connectivity for the three most porous patterns while the less porous designs had much lower connectivity (1.4% and 4.1% area fraction) concurrent with their decreased porosity and pore size.

These large, curved shapes show step/staircase artifacts (particularly in the zygoma example) because there are multiple print levels for a single level of input voxels (input voxel edge = 0.600 mm, slicing voxel edge = 0.100 mm, printing layer height = 0.200 mm). This staircase artifact could be resolved by smoothing the surface of the slicing design 3D-matrix.

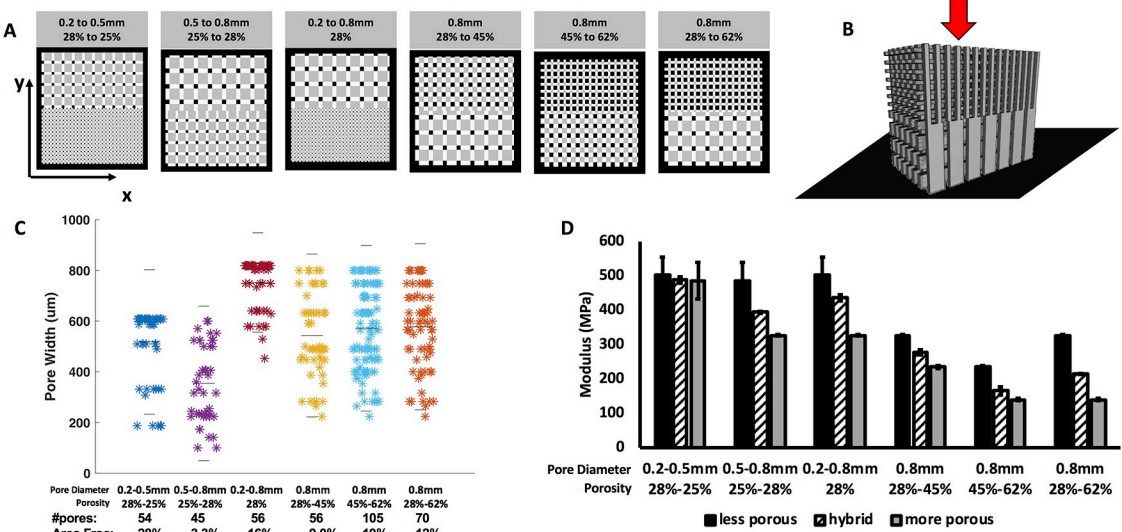

**Fig 4. 3D-printed scaffolds with hybrid pore structures.** (**A**) 3D previews of scaffold designs featuring a more porous region and less porous region which meet at a center boundary. View is top-down onto the xy surface of the scaffold (**B**) Schematic showing application of force (red arrow) and alignment of scaffold on the platen (black plane) (**C**) Pore connectivity of transition plane: measured pore areas, number of pores, and area fraction of boundary plane that is connected pore space. Gray lines indicate median and upper and lower quartiles. (**D**) The compressive modulus of each transition scaffold compared to homogenous scaffolds composed of one of the pore diameter-porosity combinations found in the transition scaffold.

# Discussion

This work develops and implements an approach for the design and manufacture of 3D-printed scaffolds for tissue engineering applications. scafSLICR provides the ability to easily leverage the available design and manufacturing space available in additive manufacturing. In

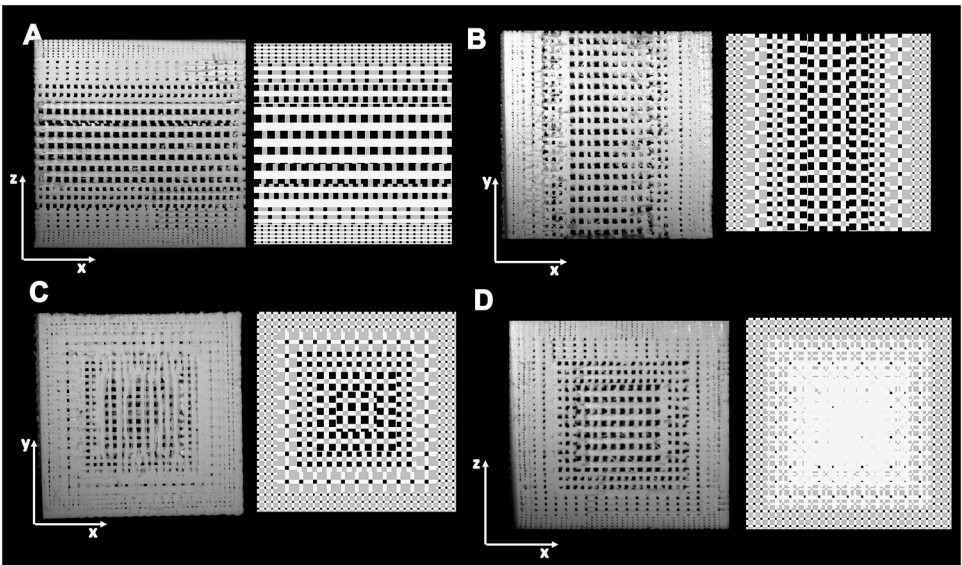

**Fig 5. 3D-printed scaffolds with heterogeneous pore structures.** Pictures of cross-sections of $2 \times 2 \times 2$ cm$^3$ ABS scaffolds (left) and design (right). (**A**) Graded in z. (**B**) Graded in xy. (**C**) Graded in xyz.

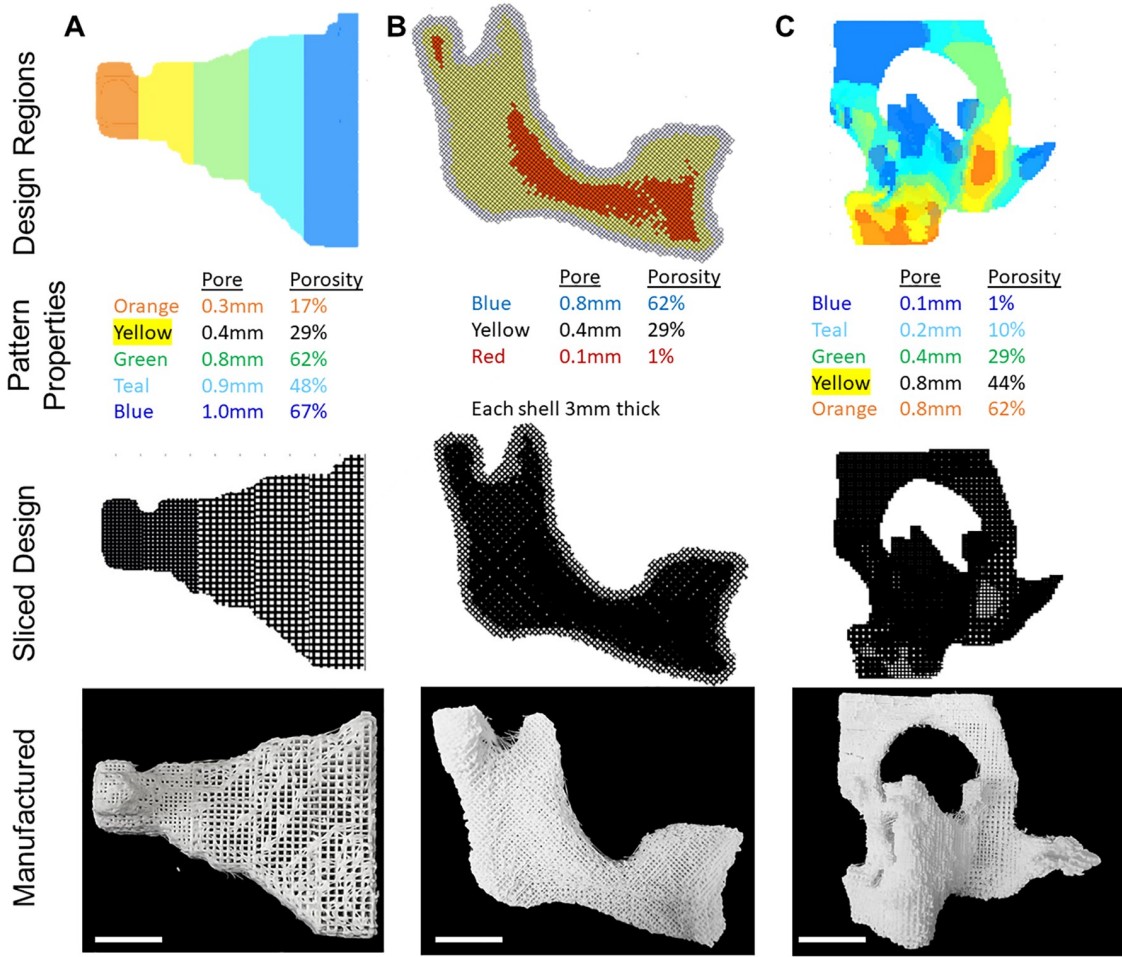

**Fig 6. 3D-printed anatomically shaped scaffolds with heterogeneous pore structures.** Anatomic shapes from the craniofacial skeleton were labeled with different design regions, sliced with scafSLICR, and 3D-printed. **(A)** Zygomatic arch patterned linearly left-to-right. **(B)** Hemi-mandible patterned with shells from exterior to interior. **(C)** Orbital midface complex patterned according to average shape thickness. Scale bar = 1cm.

particular, the broad subset of porous microarchitectures can be dependably mixed together in patterns with mechanical integrity and porous interconnectivity.

The cubic cross-hatch pore pattern used in scafSLICR has been used broadly throughout 3D-printing applications to create tissue-engineered scaffolds. We provide complexity to this structure by changing both fiber-fiber spacing and the width and height of struts *via* adjacent and stacked fibers. scafSLICR operates across the design space of the hardware (nozzle diameter) to create porous micropatterns according to desired features (pore size, porosity). This design approach permits multiple porosities with the same pore size and multiple pore sizes with the same porosity. The availability of this breadth in the design space is important because porosity is most directly attuned with print mechanics and pore size with biologic function. Thus, by decoupling the pore size and overall porosity, we have increased the versatility of the application to control the porous microarchitecture.

Different strut patterns beyond the classic cross-hatched rectangular patterns are possible. By off-setting the print direction to different angles or curves, the base pattern can be drastically alternated by *z*-layer and *xy*-location to create more complex patterns. Changing the base

design from regular cubic cross-hatched struts to another with different offset angles or arching fibers would increase the design space further, and perhaps influence mechanics and porosity in beneficial ways. For example, Moroni *et al.* manufactured scaffolds with 0-45-90 degree patterns of strut offset in order to closely match scaffold mechanics to the cartilage microenvironment [25]. Additionally, Szojka *et al.* 3D-printed scaffolds with alternating layers of parallel fibers with layers of radial ring fiber pattern [37]. Such changes could be implemented into scafSLICR by changing the template creation sub-routine. Unit cell libraries [15,38] of pore and strut shapes for 3D-printed tissue-engineered scaffolds are common for AM techniques such as SLS and powder binding, but the complex strut architecture is infeasible in fused deposition manufacturing methods.

The design limits of the microarchitectures of this study are well suited to bone tissue engineering. There are well-established constraints for bone tissue engineering scaffolds with regards to porosity, pore size, and mechanics. Porosity should be greater than 50% to provide space for tissue growth and regeneration [39–41]. Pores should range between 100μm and 1mm [39,42,43]. Mechanical moduli needed to mimic bone vary from 14 MPa of trabecular bone to 2 GPa of cortical bone [11,39,44,45].

Scaling the manufacture of unique porous architectures to large shapes has been a challenge using FDM, limiting the applications of FDM-based 3D-printing of anatomically shaped scaffolds. Many studies establish their techniques at scales less than 2 cm in regular cubes and cylinders, which facilitates insights into material properties and enables a greater understanding of the cell-material interactions. In this study, we consider challenges associated with scale-up of 3D-printed scaffolds, such as load-bearing bones, which need to be addressed to facilitate long-term clinical applications. The complex geometric nature (curves, gaps, peaks, and small walls and divots) of anatomic shapes challenges the 3D-printing processes developed for cubes and cylinders. Moreover, when developing a slicing system for tissue engineering scaffolds, it is essential that the system can readily adapt to a variety of complex anatomic shapes. scafSLICR easily scaled to large prints, with regional heterogeneity that did not compromise porous or mechanical interconnectivity.

One of the major weaknesses of this study is the choice of material. While it is bioinert, ABS was used because of manufacturing simplicity, speed, and cost. The mechanical assessments were used to validate that porosity influenced mechanics and they were not intended to demonstrate appropriateness for bone scaffold implantation. Towards that end, our research group has used scafSLICR to design and manufacture scaffolds in polycaprolactone and bioactive variations thereof (data not shown). While the examples demonstrate the applicability of 3D-printing for bone tissue engineering, many other tissue engineering applications require porous scaffolds with known pore structures and mechanics [46]. Additionally, food 3D-printing primarily uses extrusion-based printing methods, and scafSLICR could be adapted to these printing systems, enabling different porous and mechanical regions in the food structure [47,48]. Drug 3D-printing has become popular [49], but the application scales are too small for scafSLICR to be useful.

The outputs of scafSLICR enable design validation *in silico* before proceeding to manufacture or implantation. The manufactured scaffolds in this study precisely matched the porous designs. These digital porous models of the scaffolds could be used to assess properties such as mechanics, diffusion, or degradation. Such properties are difficult to directly measure, particularly in complex anatomic shapes [50–52]. The ability to validate such critical attributes are within desired ranges before manufacturing or implantation provides a low-cost means to assure implant functionality [53]. Despite the validation of the print quality and print accuracy, scafSLICR is not validated at the level needed for medical software. It would need additional dimensional and resolution tests to demonstrate reliability with many complex shapes, design

transitions, and materials. Importantly, the software validation can be compromised by the resolution and calibration of the specific 3D-printer and its ability to properly execute the GCODE. At a base-level, scafSLICR operates on a volumetric 3D-matrix. This matrix approach can be memory intensive (design matrix variables sometimes reach 5GB) but allows for the inclusion of more spatial specific information across the 3D-shape [54]. This volumetric matrix could include additional functional information (explicitly and implicitly [55])—for example, fixation attachments, or bone-implant interfaces—to improve computer-aided implant design and slicing process. Additionally, the 3D-matrix has a direct correlation to the DICOM format used to obtain patient-specific anatomic shapes and allows for minimal manipulation of that data along the design and manufacturing process. In contrast, many slicing software systems operate on the common STL format, which only includes information on the surface topography and therefore slice based on 2D contours of the design. Breaking away from the STL format to more of a computer aided design format echoing many of the thoughts from https://www.fabbaloo.com/blog/2019/2/25/at-last-the-end-of-stl-is-in-sight.

## Conclusion

This work developed an approach to designing and manufacturing 3D-printing scaffolds for tissue engineering, with direct control over scaffold features. It was successfully implemented in MATLAB (or the open-source OCTAVE) and is available at Mathworks Repository as a modifiable source code and as a user-friendly graphical user interface. Scaffolds manufactured with the approach were validated with sliced designs. Complex designs of graded pore patterns were demonstrated in regular cubes and complex anatomic shapes at scale. scafSLICR provides both an approach to designing tissue engineering scaffolds with controlled, heterogeneous complexity and scale as well as a readily available tool for tissue engineers to use in designing and manufacturing scaffolds across a variety of 3D-printing systems.

## Supporting information

**S1 Appendix. scafSLICR user guide.**
(DOCX)

**S2 Appendix. Methods to design the scaffolds used in the manuscript.**
(DOCX)

## Author Contributions

**Conceptualization:** Ethan Nyberg, Aine O'Sullivan, Warren Grayson.

**Formal analysis:** Ethan Nyberg, Aine O'Sullivan.

**Funding acquisition:** Warren Grayson.

**Methodology:** Ethan Nyberg, Aine O'Sullivan.

**Writing – original draft:** Ethan Nyberg, Aine O'Sullivan.

**Writing – review & editing:** Warren Grayson.

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
