## [Decision Letter · Decision Letter 0]

22 Aug 2019

PONE-D-19-19975

scafSLICR: a MATLAB-based Slicing Algorithm to Enable Fused Deposition Modeling 3D-Printing of Tissue Engineering Scaffolds with Heterogenous Porous Microarchitecture

PLOS ONE

Dear Dr. Grayson,

Thank you for submitting your manuscript to PLOS ONE. After careful consideration, we feel that it has merit but does not fully meet PLOS ONE’s publication criteria as it currently stands. Therefore, we invite you to submit a revised version of the manuscript that addresses the points raised during the review process.

ACADEMIC EDITOR: Please insert comments here and delete this placeholder text when finished. Be sure to:

Indicate which changes are required versus recommended for acceptanceAddress any conflicts between the reviewsProvide specific feedback from your evaluation of the manuscript

We would appreciate receiving your revised manuscript by Oct 06 2019 11:59PM. To enhance the reproducibility of your results, we recommend that if applicable you deposit your laboratory protocols in protocols.io, where a protocol can be assigned its own identifier (DOI) such that it can be cited independently in the future. For instructions see: http://journals.plos.org/plosone/s/submission-guidelines#loc-laboratory-protocols

We look forward to receiving your revised manuscript.

Kind regards,

Vahid Serpooshan, PhD

Academic Editor

PLOS ONE

Journal Requirements:

Additional Editor Comments (if provided):

Reviewers' comments:

Reviewer's Responses to Questions

**Comments to the Author**

1. Is the manuscript technically sound, and do the data support the conclusions?

Reviewer #1: Yes

Reviewer #2: Yes

2. Has the statistical analysis been performed appropriately and rigorously? 

Reviewer #1: Yes

Reviewer #2: Yes

3. Have the authors made all data underlying the findings in their manuscript fully available?

Reviewer #1: Yes

Reviewer #2: Yes

4. Is the manuscript presented in an intelligible fashion and written in standard English?

Reviewer #1: Yes

Reviewer #2: Yes

5. Review Comments to the Author

Reviewer #1: This research is very useful and practical. The reviewer thanks the authors for developing this free and open source software. It is a great contribution to scaffold design. However, the manuscript may need further improvement.

In the title, “Fused Deposition Modelling 3D Printing” is wordy, suggest removing FDM, just use 3D Printing.

The literature search on scaffold design needs a lot of improvement. Please look at Section 2 of the most recent review for the landscape of scaffold design for tissue engineering:

• Yang Y, Wang G, Liang H, et al., 2019, Additive manufacturing of bone scaffolds. Int J Bioprint, 5(1): 148

Also, when reviewing slicing methods for FDM, the following relevant works are missing:

• Adams, D. and Turner, C.J., 2018. An implicit slicing method for additive manufacturing processes. Virtual and Physical Prototyping, 13(1), pp.2-7.

• Tamburrino, F., Graziosi, S. and Bordegoni, M., 2019. The influence of slicing parameters on the multi-material adhesion mechanisms of FDM printed parts: an exploratory study. Virtual and Physical Prototyping, pp.1-17.

Equation 1, why is the right side of the equation not to the power of 3? What is the basis for using equation 1 to calculate porosity? FMD is able to print filament-based open-porous structures (layers of filament lines are stacked at different angles) as well as unit cell/strut-based open-porous structures (grid patterns of filaments are stacked with vertical struts), see works below.

• Cheah, C.M., Chua, C.K., Leong, K.F., Cheong, C.H. and Naing, M.W., 2004. Automatic algorithm for generating complex polyhedral scaffold structures for tissue engineering. Tissue engineering, 10(3-4), pp.595-610.

• Cheah, C.M., Chua, C.K., Leong, K.F. and Chua, S.W., 2003. Development of a tissue engineering scaffold structure library for rapid prototyping. Part 1: investigation and classification. The International Journal of Advanced Manufacturing Technology, 21(4), pp.291-301.

• Cheah, C.M., Chua, C.K., Leong, K.F. and Chua, S.W., 2003. Development of a tissue engineering scaffold structure library for rapid prototyping. Part 2: parametric library and assembly program. The International Journal of Advanced Manufacturing Technology, 21(4), pp.302-312.

Could the authors provide information on the time aspect of their algorithm and printing process?

The authors’ work may benefit not only tissue engineering but also related fields such as food printing and drug printing (see below recent reviews). The authors should highlight this potential in the discussion.

• Voon, S.L., An, J., Wong, G., Zhang, Y. and Chua, C.K., 2019. 3D food printing: a categorised review of inks and their development. Virtual and Physical Prototyping, 14(3), pp.203-218.

• Lepowsky, E. and Tasoglu, S., 2018. 3D printing for drug manufacturing: A perspective on the future of pharmaceuticals. Int J Bioprint, 4(1):119.

• Tan C, Toh W Y, Wong G, et al., 2018, Extrusion-based 3D food printing – Materials and machines. Int J Bioprint, 4(2): 143.

Reviewer #2: The paper presents and interesting method in a clear and professional way. The literature is fairly reviewed. There are some minor comments that should be addressed, as detailed below.

The section “Scaffold Design with MATLAB Script” should be clarified. The terms fibre and strut should be explicitly defined in the methods section, before the terms are used. It should be discussed at some point that the pores are actually long rectangular channel. The “square” nature of pores is simply their visual appearance when viewed top-down. It should be explained that equation 1 is the designed porosity (not measured porosity) because it seems that it is not used to calculate the experimental porosity (the DiameterJ plug-in is used). In my opinion, the porosity measured by measuring the square pore size is not directly comparable to the mass porosity (due to the pores actually being long channels). However, it seems that equation 1 is directly comparable to equation 2. Can the authors confirm whether the porosity values measured by mass or by optical microscopy are theoretically comparable? Many tissue engineering scaffold publications consider the area of pores based on measurements of the square-pore-area, so it is not necessarily a problem for this paper to do so. But it would be useful to highlight to the interested reader the limitation of measuring porosity based on top-down views (and also side views in the case of this publications).

Explain how the 3D models of the designs were generated in MATLAB. E.g. were the fibres assumed to be rectangular? How was the 3D model exported and analysed?

Fig 4C is not clear. The visual quality is very poor in the version supplied to me, but even without that issue, it is not clear. Is Fig 4A a top view or side view? What are the grey lines in Fig 4C? How did you analyse the models to generate the data for Fig 4C? The following sentence sounds like it is not telling the whole story: “Because the nature of the interface depends on the position, extent, and curvature of the interface surface, pattern-to-pattern interconnectivity could not be assessed experimentally and was instead measured using in silico designs of the presented examples.” … (isn’t the interface a straight surface, with a known position, extending through the model? And it seems from Fig 5A/B that a region near the interface could be analysed experimentally.) Please clarify.

Scale bars in Fig 6 should have their length indicated. Scale bars should also be included in other figures.

The statement “Many studies establish their techniques at scales less than 2 cm in regular cubes and cylinders, which poorly reflects the challenges tissue engineering seeks to address” is misleading. There are many applications for tissue engineering at the scale of <2cm. Furthermore, many of the key challenges related to tissue engineering are due to lack of fundamental understanding. Small structures are much more suitable for the development of fundamental understanding (especially when there are many inter-related and as-yet poorly understood factors). If all tissue engineering research had been completed using only large anatomical geometries, progress would be orders of magnitude slower due to the difficulty in manufacturing and characterisation. The present study is a good example – simple structures are used to generate almost all of the data (Figs 2-5) and the more complex structures (Fig 6) have very little quantitative characterisation. The authors should give a fair appraisal of the strengths of their study (longer term clinical application, etc.) and the strengths of research using simpler structures (research and shorter term clinical trials, etc.).

With regards to the limitation of studying ABS, it is good that the authors refer to use of the software for polycaprolactone. It may be the case that nothing preventing the software from being used for any material. If so, it should be stated explicitly that the software was not developed specifically for ABS and the method could, in principle, be used for any material (with appropriate calibration/optimisation of the setup parameters).

Some typographic/formatting errors:

• Page 12: “Federate” in the sentence “The federate was set to 1200 mm/min and a 1% over-extrusion factor was applied throughout the entire scaffold”.

• Page 13: The website link “www.mathworks.com/example” should be updated.

• There is no clear difference between heading style for section heading and subsection headings. This must be corrected during final formatting of the published article.

• Page 18: use 3 sig fig for all of the moduli (503, 486, 327.5). Having a 4 sig fig for one but not others is strange.

6. PLOS authors have the option to publish the peer review history of their article (what does this mean?). If published, this will include your full peer review and any attached files.

Reviewer #1: No

Reviewer #2: No

---

## [Author Response · Author response to Decision Letter 0]

19 Oct 2019

The authors would like to thank the reviewers for their insightful review of the manuscript and for their helpful comments. We have revised the manuscript based on these suggestions. These changes are highlighted in red font in the revised manuscript. 

Reviewer #1: 

This research is very useful and practical. The reviewer thanks the authors for developing this free and open source software. It is a great contribution to scaffold design. However, the manuscript may need further improvement.

1. In the title, “Fused Deposition Modelling 3D Printing” is wordy, suggest removing FDM, just use 3D Printing.

We have simplified the title as suggested to only refer to 3D-printing. 

New Title: “scafSLICR: a MATLAB-based Slicing Algorithm to Enable 3D-Printing of Tissue Engineering Scaffolds with Heterogeneous Porous Microarchitecture”

2. The literature search on scaffold design needs a lot of improvement. Please look at Section 2 of the most recent review for the landscape of scaffold design for tissue engineering:

• Yang Y, Wang G, Liang H, et al., 2019, Additive manufacturing of bone scaffolds. Int J Bioprint, 5(1): 148

We have expanded the literature search to include the manuscript cited by the reviewer and additional designs relevant to fused-deposition manufacturing in novel ways. However, we specifically focused on FDM in the Introduction and many unit cell design approaches described in the review are not easily translated to FDM systems due to resolution limitations. We included the following text:

“While there have been design approaches for selective laser sintering 3D-printing using unit cell libraries (1), topology optimization (2,3), and mathematical design (4), the structures are not reasonably transformed to the fiber deposition paradigm of FDM (5).” 

3. Also, when reviewing slicing methods for FDM, the following relevant works are missing:

• Adams, D. and Turner, C.J., 2018. An implicit slicing method for additive manufacturing processes. Virtual and Physical Prototyping, 13(1), pp.2-7.

• Tamburrino, F., Graziosi, S. and Bordegoni, M., 2019. The influence of slicing parameters on the multi-material adhesion mechanisms of FDM printed parts: an exploratory study. Virtual and Physical Prototyping, pp.1-17.

We thank the reviewer for bringing these works to our attention. We have included them in the text as follows: 

“This volumetric matrix could include additional functional information (explicitly and implicitly (6))—for example, fixation attachments, or bone-implant interfaces—to improve computer-aided implant design and slicing process.

The slicing process determines the fiber laydown pattern, and the resulting geometric and mechanical properties of the print (7).” 

4. Equation 1, why is the right side of the equation not to the power of 3? What is the basis for using equation 1 to calculate porosity? 

We clarify that we report on isotropic pore structures based on the repeating unit of any of the scaffolds. For example, in the figure below, the unit volume of the repeating unit is given by:

Unit volume = Length * width * height

 = (Strut width + Pore width)*(strut width + Pore width) * Pore height

The porous void of the unit volume:

 = Pore width * Pore width * Pore height

The porosity = (Pore width * Pore width * Pore height)/((Strut width + Pore width)*(Strut width + Pore width) * Pore height)

 = (Pore width)2/(Strut width + Pore width)2

To better explain this, we have included the following text:

“The porosity of the repeating unit is calculated in equation 1: a repeating unit has the width of a strut + pore, the length of a strut + pore, and the height of pore, while the void volume is given by pore width*pore width*pore height.” 

5. FDM is able to print filament-based open-porous structures (layers of filament lines are stacked at different angles) as well as unit cell/strut-based open-porous structures (grid patterns of filaments are stacked with vertical struts), see works below.

• Cheah, C.M., Chua, C.K., Leong, K.F., Cheong, C.H. and Naing, M.W., 2004. Automatic algorithm for generating complex polyhedral scaffold structures for tissue engineering. Tissue engineering, 10(3-4), pp.595-610.

• Cheah, C.M., Chua, C.K., Leong, K.F. and Chua, S.W., 2003. Development of a tissue engineering scaffold structure library for rapid prototyping. Part 1: investigation and classification. The International Journal of Advanced Manufacturing Technology, 21(4), pp.291-301.

• Cheah, C.M., Chua, C.K., Leong, K.F. and Chua, S.W., 2003. Development of a tissue engineering scaffold structure library for rapid prototyping. Part 2: parametric library and assembly program. The International Journal of Advanced Manufacturing Technology, 21(4), pp.302-312.

We thank the reviewer for bringing these publications to our attention. We have cited them in our discussion of open-porous structures based on fiber angle and have indicated the difficulty of using unit cell libraries in FDM. 

“Unit cell libraries(1,8) for 3D-printed tissue-engineered scaffolds are common for AM techniques such as SLS and powder binding, but the complex strut architecture is infeasible in fused deposition manufacturing methods.” 

6. Could the authors provide information on the time aspect of their algorithm and printing process?

We indicate in the results section that the largest shape required 6 minutes of computational time to prepare gcode. All smaller shapes required proportionally less time as the algorithm workload varies directly with the volume of the shape. Printing time similarly depended on the volume of the object and ranged from 15 min for 1cm cubes to ~10 hours for the large anatomic objects. We have included the following text to indicate that the algorithm and print time scales with the volume of the object. 

“Smaller volume shapes required proportionally less time to slice. 3D-printing time similarly scaled with volume, with large shapes requiring on the order of 10 hours and smaller shapes on the order of 20 min.” 

7. The authors’ work may benefit not only tissue engineering but also related fields such as food printing and drug printing (see below recent reviews). The authors should highlight this potential in the discussion.

• Voon, S.L., An, J., Wong, G., Zhang, Y. and Chua, C.K., 2019. 3D food printing: a categorised review of inks and their development. Virtual and Physical Prototyping, 14(3), pp.203-218.

• Lepowsky, E. and Tasoglu, S., 2018. 3D printing for drug manufacturing: A perspective on the future of pharmaceuticals. Int J Bioprint, 4(1):119.

• Tan C, Toh W Y, Wong G, et al., 2018, Extrusion-based 3D food printing – Materials and machines. Int J Bioprint, 4(2): 143.

We appreciate the reviewer's consideration of food and drug printing along with the potential applications of this work to those fields. We have included the following text in the discussion:

“Food 3D-printing primarily uses extrusion-based printing methods, and scafSLICR could be adapted to these printing systems, enabling different porous and mechanical regions in the food structure (9,10). Drug 3D-printing has become popular (11), but the application scales are too small for scafSLICR to be useful.” 

 

Reviewer #2: 

The paper presents and interesting method in a clear and professional way. The literature is fairly reviewed. There are some minor comments that should be addressed, as detailed below.

1. The section “Scaffold Design with MATLAB Script” should be clarified. The terms fibre and strut should be explicitly defined in the methods section, before the terms are used. It should be discussed at some point that the pores are actually long rectangular channel. The “square” nature of pores is simply their visual appearance when viewed top-down. 

We have defined each of these terms the first time they are used in the manuscript.

“Fiber – Structure of extruded of material from the extruder head along a toolpath on a single print layer. Fibers are assumed to be rectangular with the width of the extruder nozzle and the height of the print layer. 

Strut – Solid material of many adjacent fibers, often composed of multiple fibers in width and height. 

Pore and Channel – Void spacing between struts, in horizontal and vertical directions. Square projections with equal width and height when viewed from the top or side of the scaffold.” 

2. It should be explained that equation 1 is the designed porosity (not measured porosity) because it seems that it is not used to calculate the experimental porosity (the DiameterJ plug-in is used). In my opinion, the porosity measured by measuring the square pore size is not directly comparable to the mass porosity (due to the pores actually being long channels). However, it seems that equation 1 is directly comparable to equation 2. Can the authors confirm whether the porosity values measured by mass or by optical microscopy are theoretically comparable? Many tissue engineering scaffold publications consider the area of pores based on measurements of the square-pore-area, so it is not necessarily a problem for this paper to do so. But it would be useful to highlight to the interested reader the limitation of measuring porosity based on top-down views (and also side views in the case of this publications).

We have clarified that the measured porosity in this study was done using mass measurements. We include the following statement:

“The porosity of printed porous scaffolds was determined by mass measurements compared to solid prints of the same dimensions (Equation 2), and they were compared to the porosity of the computer design by computing the void fraction the 3D-MATLAB matrix.”

3. Explain how the 3D models of the designs were generated in MATLAB. E.g. were the fibres assumed to be rectangular? How was the 3D model exported and analyzed?

We have provided clarity in the methods section by adding the following text: 

“3D-Design models were generated in MATLAB by assuming fibers to be perfectly rectangular (nozzle width x print layer height). The design was assembled into a volumetric 3D-matrix which could be examined directly in MATLAB using matrix property analysis, MATLAB 3D-plotting functions, or exported as an STL to be viewed and analyzed in a range of software programs.” 

4. Fig 4C is not clear. The visual quality is very poor in the version supplied to me, but even without that issue, it is not clear. Is Fig 4A a top view or side view? What are the grey lines in Fig 4C? How did you analyse the models to generate the data for Fig 4C? 

We thank the reviewer for this useful critique. We have edited the figure to increase clarity: (1) We have added labels to indicate the view angles. (2) We describe the gray lines as mean and quartile marks in the caption. We clarify in the text (Methods / Analysis of Porous Boundaries) that we digitally isolated the interface and measured the pore areas in that thin section of the model: 

“In order to assess the connectivity of pores between regions of different porous microarchitectures, the area of the interface was analyzed in the digital model in MATLAB. The interface was isolated digitally at a 200µm thickness, and the area of each connecting pore was measured. The porous area fraction of the boundary surface was found by summing the individual pore areas and dividing by the area of the interface boundary between regions.” 

The following sentence sounds like it is not telling the whole story: “Because the nature of the interface depends on the position, extent, and curvature of the interface surface, pattern-to-pattern interconnectivity could not be assessed experimentally and was instead measured using in silico designs of the presented examples.” … (isn’t the interface a straight surface, with a known position, extending through the model?) 

We could not get uniform cuts using any of the available saws and grinding tools to isolate the plane of the transition for experimental analysis. 

5. And it seems from Fig 5A/B that a region near the interface could be analysed experimentally. Please clarify.

While the reviewer is correct in theory, we were unable to experimentally isolate uniform regions in those scaffolds near the interface using cutting, grinding, or cracking methods. 

6. Scale bars in Fig 6 should have their length indicated. Scale bars should also be included in other figures.

We have included scale bars in all of the relevant figures.

7. The statement “Many studies establish their techniques at scales less than 2 cm in regular cubes and cylinders, which poorly reflects the challenges tissue engineering seeks to address” is misleading. There are many applications for tissue engineering at the scale of <2cm. Furthermore, many of the key challenges related to tissue engineering are due to lack of fundamental understanding. Small structures are much more suitable for the development of fundamental understanding (especially when there are many inter-related and as-yet poorly understood factors). If all tissue engineering research had been completed using only large anatomical geometries, progress would be orders of magnitude slower due to the difficulty in manufacturing and characterization. The present study is a good example – simple structures are used to generate almost all of the data (Figs 2-5) and the more complex structures (Fig 6) have very little quantitative characterization. The authors should give a fair appraisal of the strengths of their study (longer term clinical application, etc.) and the strengths of research using simpler structures (research and shorter-term clinical trials, etc.).

We thank the reviewer for this critique. We have re-written the statement to clearly indicate the relative strengths of small vs large scale 3D-printed tissue engineering scaffolds: 

“Many studies establish their techniques at scales less than 2 cm in regular cubes and cylinders, which facilitates insights into material properties and enables a greater understanding of the cell-material interactions. In this study, we consider challenges associated with scale-up of 3D-printed scaffolds, such as load-bearing bones, which need to be addressed to facilitate long-term clinical applications.” 

8. With regards to the limitation of studying ABS, it is good that the authors refer to use of the software for polycaprolactone. It may be the case that nothing preventing the software from being used for any material. If so, it should be stated explicitly that the software was not developed specifically for ABS and the method could, in principle, be used for any material (with appropriate calibration/optimization of the setup parameters).

We thank the reviewer for this suggestion. We have indicated this versatility in the methods section with the following text: 

“scafSLICR could work with a range of materials in an FDM printer, optimized with printing parameters.” 

9. Some typographic/formatting errors:

• Page 12: “Federate” in the sentence “The federate was set to 1200 mm/min and a 1% over-extrusion factor was applied throughout the entire scaffold”.

• Page 13: The website link “www.mathworks.com/example” should be updated.

• There is no clear difference between heading style for section heading and subsection headings. This must be corrected during final formatting of the published article.

• Page 18: use 3 sig fig for all of the moduli (503, 486, 327.5). Having a 4 sig fig for one but not others is strange.

We thank the reviewers for their time carefully reading the manuscript and have made the suggested edits. 

References

1. Chua CK, Leong KF, Cheah CM, Chua SW. Development of a Tissue Engineering Scaffold Structure Library for Rapid Prototyping. Part 1: Investigation and Classification. Int. J. Adv. Manuf. Technol. 21, 291, 2003; 

2. Hollister SJ, Maddox RD, Taboas JM. Optimal design and fabrication of scaffolds to mimic tissue properties and satisfy biological constraints. Biomaterials. 23(20), 4095, 2002; 

3. Challis VJ, Guest JK, Grotowski JF, Roberts AP. Computationally generated cross-property bounds for stiffness and fluid permeability using topology optimization. Int. J. Solids Struct. [Internet]. Elsevier Ltd; 49(23–24), 3397, 2012; Available from: http://dx.doi.org/10.1016/j.ijsolstr.2012.07.019

4. Yang Y, Wang G, Liang H, Gao C, Peng S, Shen L. Additive manufacturing of bone scaffolds. Int. J. Bioprinting. 0, 1, 2019; 

5. Cheah C, Chua C, Leong K, Cheong C, Naing M-W. Automatic Algorithm for Generating Complex Polyhedral Scaffold Structures for Tissue Engineering. Tissue Eng. 10(3), 2004; 

6. Adams DW, Turner CJ. IMPLICIT SLICING METHOD FOR ADDITIVE MANUFACTURING PROCESSES. Solid Free Form Fabr. 844, 2017; 

7. Tamburrino F, Graziosi S, Bordegoni M, Tamburrino F, Graziosi S. The influence of slicing parameters on the multi- material adhesion mechanisms of FDM printed parts : an exploratory study FDM printed parts : an exploratory study. Virtual Phys. Prototyp. Taylor & Francis; 2759, 2019; 

8. Kang H, Lin CY, Hollister SJ. Topology optimization of three dimensional tissue engineering scaffold architectures for prescribed bulk modulus and diffusivity. Struct. Multidiscip. Optim. 42(4), 633, 2010; 

9. Yan W, Extrusion L, Tan C, Toh WY, Wong G, Li L. Extrusion-based 3D food printing – Materials and machines. Int. J. Bioprinting. 4(2), 0, 2018; 

10. Voon SL, An J, Wong G, Zhang Y, Chua CK. 3D food printing: a categorised review of inks and their development. Virtual Phys. Prototyp. Taylor & Francis; 14(3), 201, 2019; 

11. Lepowsky E, Tasoglu S. 3D printing for drug manufacturing: A perspective on the future of pharmaceuticals. Int. J. Bioprinting. 4(1), 1, 2018;

---

## [Decision Letter · Decision Letter 1]

28 Oct 2019

scafSLICR: a MATLAB-based Slicing Algorithm to Enable  3D-Printing of Tissue Engineering Scaffolds with Heterogenous Porous Microarchitecture

PONE-D-19-19975R1

Dear Dr. Grayson,

We are pleased to inform you that your manuscript has been judged scientifically suitable for publication and will be formally accepted for publication once it complies with all outstanding technical requirements.

With kind regards,

Vahid Serpooshan, PhD

Academic Editor

PLOS ONE

Additional Editor Comments (optional):

Reviewers' comments:

Reviewer's Responses to Questions

**Comments to the Author**

1. If the authors have adequately addressed your comments raised in a previous round of review and you feel that this manuscript is now acceptable for publication, you may indicate that here to bypass the “Comments to the Author” section, enter your conflict of interest statement in the “Confidential to Editor” section, and submit your "Accept" recommendation.

Reviewer #1: All comments have been addressed

Reviewer #2: All comments have been addressed

2. Is the manuscript technically sound, and do the data support the conclusions?

Reviewer #1: Yes

Reviewer #2: Yes

3. Has the statistical analysis been performed appropriately and rigorously? 

Reviewer #1: Yes

Reviewer #2: Yes

4. Have the authors made all data underlying the findings in their manuscript fully available?

Reviewer #1: Yes

Reviewer #2: Yes

5. Is the manuscript presented in an intelligible fashion and written in standard English?

Reviewer #1: Yes

Reviewer #2: Yes

6. Review Comments to the Author

Reviewer #1: All my comments have been properly addressed. The revision is satisfactory. I recommend it for publication now.

Reviewer #2: The manuscript has been improved and the recommendations for revision have been effectively addressed or rebutted.

7. PLOS authors have the option to publish the peer review history of their article (what does this mean?). If published, this will include your full peer review and any attached files.

Reviewer #1: No

Reviewer #2: No

---

## [Editor Report · Acceptance letter]

8 Nov 2019

PONE-D-19-19975R1 

scafSLICR: a MATLAB-based Slicing Algorithm to Enable 3D-Printing of Tissue Engineering Scaffolds with Heterogeneous Porous Microarchitecture 

Dear Dr. Grayson:

I am pleased to inform you that your manuscript has been deemed suitable for publication in PLOS ONE. Congratulations! Your manuscript is now with our production department. 

With kind regards,

on behalf of

Dr. Vahid Serpooshan 

Academic Editor

PLOS ONE